# SITUATED COMMUNICATION: A SOLUTION TO OVER-COMMUNICATION BETWEEN ARTIFICIAL AGENTS

**Aleksandra Kalinowska**
Northwestern University, Evanston, IL
& DeepMind, Edmonton, AB
ola@u.northwestern.edu

**Elnaz Davoodi**
DeepMind
Montreal, QC
elnazd@deepmind.com

**Florian Strub**
DeepMind
Paris, France
fstrub@deepmind.com

**Kory W. Mathewson**
DeepMind
Montreal, QC
korymath@deepmind.com

**Todd D. Murphey**
Northwestern University
Evanston, IL
t-murphey@northwestern.edu

**Patrick M. Pilarski**
University of Alberta, Edmonton, AB
& DeepMind, Edmonton, AB
ppilarski@deepmind.com

## ABSTRACT

Most research on communication emergence between reinforcement learning (RL) agents explores *unsituated* communication in one-step referential tasks. The tasks are not temporally interactive and lack time pressures typically present in natural communication and language learning. In these settings, agents can successfully learn to communicate, but they do not learn to exchange information concisely—they tend towards over-communication and an anti-efficient encoding. In our work, we introduce *situated* communication by imposing an opportunity cost on communication—the acting agent has to forgo an action to solicit information from its advisor. *Situated* communication mimics the *external* pressure of passing time in real-world communication. We compare language emergence under this pressure against language learning with an *internal* cost on articulation, implemented as a per-message penalty. We find that while both pressures can disincentivise over-communication, *situated* communication does it more effectively and, unlike the internal pressure, does not negatively impact communication emergence. Implementing an opportunity cost on communication might be key to shaping language properties and incentivising concise information sharing between artificial agents.

## 1 INTRODUCTION

Effective communication is a key skill for collaboration in a multi-agent setting. As humans, we share communication protocols and cooperative conventions that have evolved over thousands of generations to optimize communication efficiency. As an example, we communicate in accordance with cooperative principles, such as Grice's maxims of conversation (Grice, 1975). In line with the Gricean maxim of quantity, we are known to try to be as informative as possible, giving only as much information as is needed (Grice, 1975). If future artificial systems are to cooperate with humans, it will be beneficial for their communication protocols to follow these patterns (Crandall et al., 2018; Steels, 2003). As a result, understanding the process of communication emergence and the pressures that shape the emergent communication protocols is of interest to the scientific community.

With a recent increase in available computational power, the field has seen a lot of progress (Wagner et al., 2003; Lazaridou & Baroni, 2020). Thus far, emergent communication between reinforcement learning (RL) agents has largely been studied in one-step referential games, such as the Lewis signalling task (Li & Bowling, 2019; Lazaridou et al., 2018; Chaabouni et al., 2022). This type

of learning environment is known to successfully enable language development (Kirby & Hurford, 2002). However, prior work shows that the emerged communication protocols often do not share properties of natural languages (Kottur et al., 2017) and that artificial agents tend towards an anti-efficient encoding (Chaabouni et al., 2019). This likely happens because in the Lewis signalling task, as well as in other simulated environments (Cao et al., 2018; Kajić et al., 2020), the communication is not *situated* in the task—there is no opportunity cost to communication. In this setting, excessive use of communication does not negatively affect the outcome of the game (or cause agent frustration), as it might in a real-world situation (Steels & Brooks, 1995). Agents are not incetivised to be efficient in conveying information.

In this work, we explore internal and external pressures that can incentivise conciseness during communication emergence. As our testbed, we use a cooperative multi-step navigation task with two RL agents. In the task, a speaker provides navigation hints to help a listener reach a goal within a grid-world maze. We explore three training regimes: (i) *unsituated* communication; the speaker sends a message to the listener at each timestep without any cost, similar to the communication paradigm in existing work (Chaabouni et al., 2019; Li & Bowling, 2019; Lazaridou et al., 2018). (ii) *unsituated* communication with a per-message penalty; the speaker experiences a cost on communication effort, (iii) *situated* communication; the listener has to forgo an action to solicit information from the advising agent, experiencing an opportunity cost on communication. Using the collaborative navigation task, we evaluate how these pressures can incentivise sparse information sharing during communication emergence. We find that *situated* communication outperforms an internal cost on communication effort in terms of both conciseness of the emerged communication protocol and overall task performance.

## 2 EXPERIMENTAL SETUP

**The environment.** We define a cooperative navigation task as a Markov Decision Process (MDP) with two RL agents. The environment is set up as a pixel-based gridworld (7 by 7 cells) with a maze inside. The tested maze is made up of three T-junctions, each allowing a right and left turn. Features of the world, which are encoded with binary vectors, are represented with colors: walls are black, the maze is white, the agent is green, and the target is blue. See Fig. 1 for a visualization.

**The agents.** There are two independent RL agents, a speaker and a listener (i.e. acting agent). The speaker does not reside within the gridworld and cannot take environmental actions (i.e. navigate the maze) but instead can communicate information to the listener. The speaker's action (i.e. message) space spans 5 symbols $[0, 1, ..., 4]$. We refer to the message $m_t = 0$ as a null message; we refer to messages $m_t \in [1, 2, 3, 4]$ as non-zero messages. At each timestep, the speaker can see the entire gridworld, including the location of the agent and the location of the goal. The speaker's view of the world map is rotated to align with the direction that the listener is facing. The listener is embedded inside the gridworld and can take actions to move through the maze. The action space of the listener spans 5 actions [move up, move down, move right, move left, stay in place]. The listener's observation consists of the environmental view (if any) concatenated with the message from the speaker. We test the listener under two conditions: (1) with no visibility, where the listener's observation consists solely of the speaker's message, and (2) with partial visibility, where the listener can see the 3 pixels directly in front of them. The second variant gives the listener some environmental context to take actions without needing to rely solely on communication. We test agents with and without memory. Agents without memory have to rely only on their current observations to take an action. Agents with memory have an internal representation of the history of an episode—they can use accumulated knowledge from prior timesteps to make decisions in the current timestep. See Appendix Section A.1 for implementation details.

**The task.** The goal of the agents is to cooperate so that the listener reaches the target. In each experimental episode, both agents receive a reward $R = 1$ if the listener reaches the target before the episode terminates. Episode timeout is set to 100 steps. The goal locations are randomly assigned to one of 4 corners in the T-maze, as indicated with stars in Fig. 1. In each episode, the listener agent starts from the bottom middle cell.

**Communication modes.** We compare two modes of communication: (1) *unsituated* real-time messaging—the communication is free and guaranteed to the agent at every timestep, and (2) *situated* communication—the acting agent can actively choose between (i) taking an environmental action and (ii) soliciting information from the speaker. In *unsituated* communication, the speaker

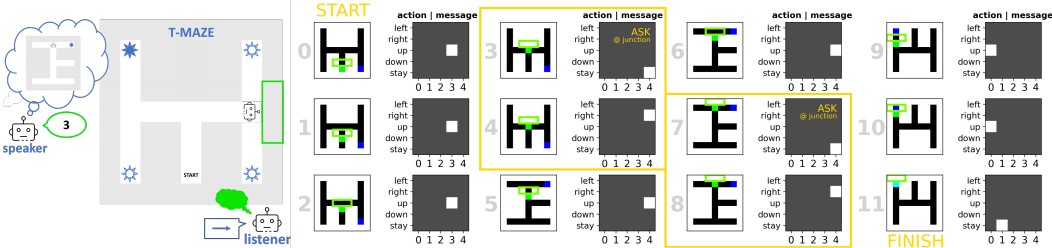

Figure 1: Experimental setup and a walk-through of an example episode with *situated* communication.

generates a 1-token message at every timestep and the message gets broadcasted to the listener before they choose an action. We refer to the listener using this mode as a passive listener, because it passively receives the speaker's message at every timestep. In *situated* communication, the message is only broadcasted to the listener after they ask for information—we refer to the listener as active. The active listener can solicit to receive information in the next timestep by choosing a *"stay in place"* action at the current timestep. The active listener experiences an opportunity cost to communication—they have to forego an environmental move (that could bring them closer to the target) in order to obtain information from the speaker and make an informed decision. As a result, they have to learn *whether* to communicate at all.

**Message penalties.** In *unsituated* communication, we test the consequences of introducing a cost of articulation for the speaker. In the baseline scenario, all messages are free to the agent. With a cost of articulation, each non-zero message incurs a penalty, while a null message (the symbol $0$) remains free to the agent. The message penalty remains constant throughout the entire duration of training. We test fixed penalties with values $m_p = [0.0, 0.01, 0.05, 0.1]$. In Figure 2, we present results with $m_p = 0.01$. For other results, refer to the Appendix.

**Evaluation metrics.** We evaluate agent performance according to three criteria: (1) task success (via the mean return per episode), (2) solution optimality (via the normalized reward per step), and (3) communication efficiency (via communication sparsity). The metric of task success, calculated as $(\sum_t R_{current\ episode})/n_{episodes}$, represents the likelihood of the agents succeeding at reaching the target before episode timeout. When it converges to $1$, agents are reliably reaching the target in each episode. The normalized reward per step quantifies the optimality of the path taken to solve the task. If the task is solved in the optimal number of steps ($n_{opt\ steps} = 9$), agents obtain a per-step reward of $1$. Formally, it is calculated as $n_{opt\ steps} * (R_{current\ episode}/n_{episode\ steps})$. Finally, the metric of communication sparsity quantifies the efficiency of information exchanged between the collaborating agents. We define it as the negative log of non-zero messages generated by the speaker per episode, such that $m_s = -\log(\sum_t \mathbb{1}_{m_t \neq 0})/n_{episodes}$ where $m_t$ is the message per timestep. Agent pairs that converge to a higher value form a more efficient communication protocol—they communicate fewer messages per episode. If agents were able to solve the task using a single message, their sparsity metric would equal $0$. Depending on the listener's characteristics: partial or no visibility, the optimal number of messages is equal to two ($m_s = -0.7$) or nine ($m_s = -2.2$) messages per episode for agents without memory, and one ($m_s = 0$) message per episode for agents with memory.

## 3 RESULTS

We experimentally compare pressures that can incentivise a concise exchange of information during communication emergence. Specifically, we consider two types of pressure:

- Internal: similar to a cost of articulation. During *unsituated* communication, we impose a cost on communication effort through a penalty for non-zero messages.
- External: mimicking the pressure of time in a real-world situation. We study agents using *situated* communication, where at each step they face an action-communication trade-off.

In our experiments, we train agent pairs communicating (1) using *unsituated* real-time communication without any pressures (our baseline), (2) under an internal pressure (using *unsituated* communication with a per-message penalty), and (3) under an external pressure (using *situated* communi-

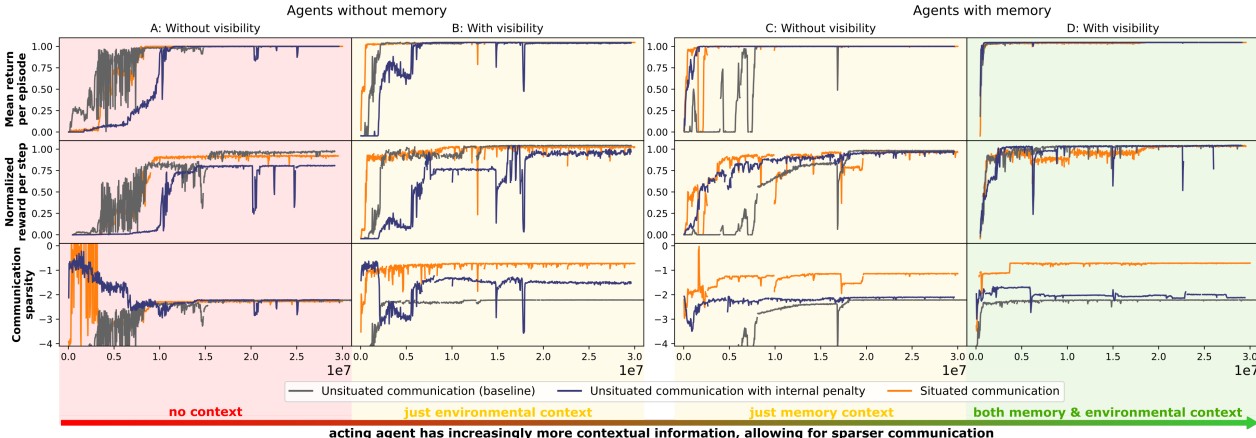

Figure 2: Comparison of pressures for avoiding over-communication (best agent pairs over 10 seeds).

cation without message penalties). In all experiments, we train agents under two visibility and two memory conditions (see Section 2 for more details). The learning curves of best agent pairs under each set of conditions are visualized in Fig. 2. Note that without memory and without visibility (column A in Fig. 2), the acting agent has no contextual information and at each timestep relies solely on the speaker's message to choose an action. With memory and/or visibility (columns B-D in Fig. 2), the listener has partial context, allowing for a possibly sparser exchange of information.

***Unsituated* communication (baseline): Without pressures to be concise, agents successfully learn to solve the task.** Baseline agents (represented with a grey line in Fig. 2) converge to an optimal solution for all tested conditions—their mean return per episode and normalized reward per step both converge to 1. However, their communication protocol is not sparse—for all tested conditions, their sparsity metric is lowest compared to agents with added pressures (refer to the grey lines in Fig. 2 row 3). It is worth noting that under the no memory and no visibility condition, all best agent pairs achieve the same sparsity—when the acting agent lacks context, agents cannot communicate more efficiently than using one message per step (refer to the bottom plot in Fig. 2 column A).

***Unsituated* communication with a penalty: A fixed penalty improves communication sparsity but makes communication emergence more difficult**. When possible, introducing a fixed penalty makes the communication sparser compared to baseline (refer to the blue vs. grey curves in Fig. 2 row 3). However, with this internal pressure, language learning becomes more difficult—introducing a fixed penalty makes finding an optimal solution harder as evidenced by the normalized reward per step not converging to 1 even for the best agent pairs (refer to row 2, columns A&B of Fig. 2). The same trends are observed while sweeping over different penalty values as detailed in Section A.2.2 of the Appendix. Lastly, note in the bottom plots in Fig. 2 that when a cost of articulation is introduced, speakers send few non-zero messages at the beginning of training. The decrease in early exploration of a common language likely makes it harder for agents to establish a successful communication protocol and consequently causes difficulty in collaborating to reach the target.

***Situated* communication: The pressure of time in a multi-step interaction can incentivise sparse communication without negatively affecting communication emergence.** In the last set of experiments, we evaluate the impact of *situated* communication on language emergence. We find that *situated* communication incentivises conciseness without stifling early exploration. Active listeners learn over time that communication comes at a cost, adjusting *when* and *whether* they solicit information, rather than avoiding communication early on (as is the case for agents with a per-message penalty). Agents using *situated* communication achieve highest communication sparsity (refer to the orange lines in Fig. 2 row 3). Importantly, the sparsity does not result in a loss in task performance—agents with an active listener reach a quasi-optimal policy (refer to Fig. 2 row 2). They follow learning patterns similar to baseline agents. Under some conditions, agents with *situated* communication converge to an optimal solution in fewer timesteps than baseline agents (refer to Fig. 2 row 2, column B&C). Overall, the external pressure of time allows agents to communicate

more concisely and find the optimal solution more quickly and more reliably than with an internal pressure on communication effort.

## 4 CONCLUSIONS & DISCUSSION

We find that giving the listener agency to choose *whether* to communicate enables agents to learn how to concisely exchange task-relevant information. By *situating* the communication in the task, we allow the functional pressure of time to shape the emergent communication protocol to be sparse, in line with the Gricean maxim of quantity. In this way, we also improve convergence to a collaborative task solution. An internal cost of articulation, implemented in the form of a per-message penalty, is not as effective.

Our results suggest that *situating* the communication in the task plays a key role in shaping the emergent communication protocol. However, the authors would like to recognize that in this work we are using a simplified version of *situated* communication to make it easier to test our hypotheses about its impact on communication emergence. In his review, Wagner et al. (2003) writes that "in nonsituated simulations an agent's actions consist solely of sending and receiving signals." On the other hand, "to be situated, an agent must also interact in noncommunicative ways with various entities such as food, predators, and other agents and must have outputs that can affect the environment and/or modify its own internal state. [...] Just being in an artificial world in which objects can be perceived is not enough for an agent to be classified as situated". In the *nonsituated* communication mode in our experiments, the speaker has only noncommunicative actions available to them (just like in Wagner et al. (2003)) whereas the listener receives signals and acts on them, allowing us to evaluate their understanding of the received messages. In the *situated* communication mode in our experiments, the speaker still has only noncommunicative actions available to them but the active listener can now choose between taking communicative and noncommunicative actions. Our ongoing work will expand this implementation and situate both the speaker and listener in the environment, enabling both agents to communicate and take noncommunicative actions in the gridworld environment.

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

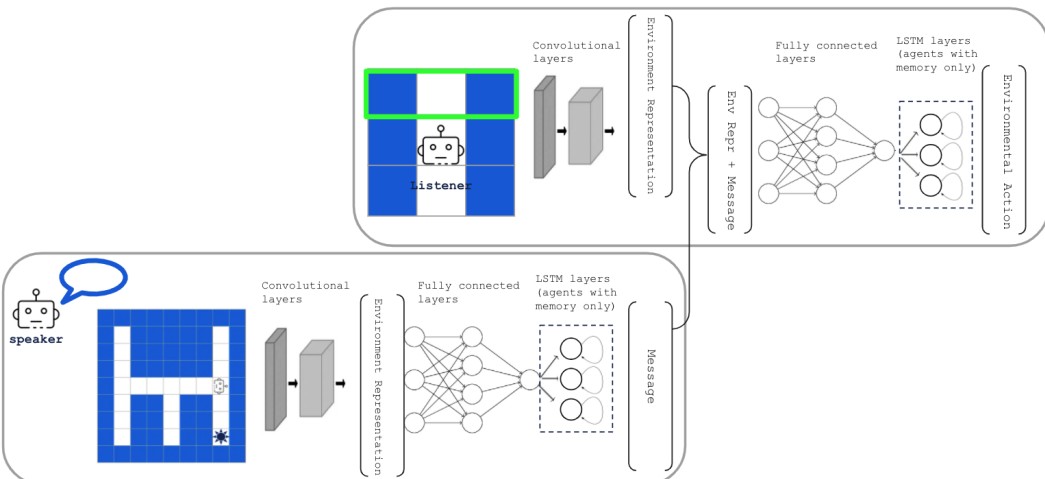

Figure 3: Agents architecture and communication dynamic.

Angeliki Lazaridou and Marco Baroni. Emergent multi-agent communication in the deep learning era. *arXiv preprint arXiv:2006.02419*, 2020.

Angeliki Lazaridou, Karl Moritz Hermann, Karl Tuyls, and Stephen Clark. Emergence of linguistic communication from referential games with symbolic and pixel input. *Int. Conf. on Learning Representations*, 2018.

Fushan Li and Michael Bowling. Ease-of-teaching and language structure from emergent communication. *Advances in Neural Information Processing Systems*, 2019.

Martin Riedmiller. Neural fitted q iteration–first experiences with a data efficient neural reinforcement learning method. *European Conf. on Machine Learning*, 2005.

Luc Steels. Evolving grounded communication for robots. *Trends in Cognitive Sciences*, 2003.

Luc Steels and Rodney Brooks. The artificial life route to artificial intelligence: Building embodied, situated agents, 1995.

Kyle Wagner, James A Reggia, Juan Uriagereka, and Gerald S Wilkinson. Progress in the simulation of emergent communication and language. *Adaptive Behavior*, 2003.

## A    APPENDIX

### A.1    IMPLEMENTATION DETAILS

**Agent architectures.** The speaker and the listener are designed as two independent RL agents. Both agents have the same architecture without sharing weights or gradient values. They both have a 2-layer Convolutional Neural Network (CNN) that generates an $8 - 32$ bit representation of the environment. In the case of the listener, this representation of the environment gets concatenated with the message received from the speaker. In both cases, the vector gets passed into a fully connected layer that generates the agent's action. Agents with memory, have an additional single-layer LSTM (Hochreiter & Schmidhuber, 1997) after their fully connected layer.

We train the agents using neural fitted Q learning (Riedmiller, 2005), with an Adam optimizer (Kingma & Ba, 2015) and $Q_t(\lambda)$ where $\lambda = 0.9$. The Q values are updated using temporal difference (TD) error where the bootstrapped $Q_t(\lambda)$ is defined as follows:

$$Q_t(\lambda) = (1 - \lambda) \sum_{n=1}^{\infty} \lambda^{n-1} Q_t^{(n)}$$

During training, agents use an $\epsilon$-greedy policy with the exploration rate set as $\epsilon = 0.01$.

**Hyperparameters.** After an initial exploration, we use a reward discount $\gamma = 0.99$. For each experiment, we run a hyperparameter sweep over learning rates of the speaker and listener $\alpha = [10^{-5}, 10^{-6}, 10^{-7}]$ and over the size of the environmental representation $s = [4, 8, 16, 32]$. We run the simulation with each hyperparameter setting 10 times with different random seeds. In the results for each experiment, we present the best performing agent pair from our hyperparameter sweep and/or the mean over the 10 replicas with the same hyperparameters as the best performing pair. The best performing agent pairs are selected based on the metric of solution optimality (the normalized reward per step). When we plot metric means, we include the standard error of the mean.

## A.2 MORE DETAILED RESULTS

### A.2.1 COMPARISON OF PRESSURES

In Fig. 2 in the main text, we visualize the best performing agents in each hyperparameter sweep. In Fig. 4 below, we visualize the agents' average performance when initialized with different random seeds (and the same hyperparameters as the best performing agent pair). The trends are similar to those described in the main text. Agents using *unsituated* communication with an internal pressure are the least likely to converge to an optimal solution. Agents using *situated* communication perform best overall. They achieve highest communication sparsity while maintaining task performance comparable to that of the baseline agents.

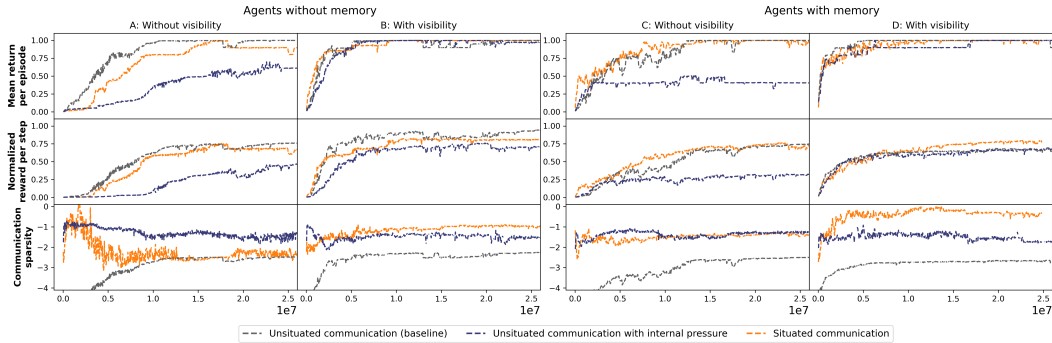

Figure 4: Comparison of pressures for avoiding over-communication (mean over 10 seeds for each experiment).

### A.2.2 UNSITUATED COMMUNICATION WITH MESSAGE PENALTIES

**To communicate efficiently, the speaker benefits from an internal penalty, mimicking communication effort. However, the additional pressure makes learning more difficult.** We find that in some cases message penalties enable the emergence of a sparse and successful communication protocol (see Fig. 5). In our penalty sweep for agents without memory, we observe two agent pairs that converge to an optimal cooperative task solution—when the speaker has a fixed message penalty of $0.05$ and the listener has no visibility, and when the speaker has a penalty of $0.01$ and communicates with a listener with partial visibility. Overall, language learning becomes more difficult due to the internal penalty on communication. Fewer agents converge and time to convergence is longer as compared to agents communicating in an environment with a free communication channel (visualized with a grey line in Fig. 5). In particular, we observe that a cost on communication stifles early exploration of a shared language. Note in the bottom plots in Fig. 5 that when a cost of articulation is introduced, speakers send almost no non-zero messages at the beginning of training. This makes it difficult for the collaborating agents to establish a successful communication protocol and consequently to succeed at the task. We also tested a pro-social reward, it did not seem to make a significant difference in any of the performance metrics.

**With memory, performance improves, but the penalty is largely ineffective while still making learning more difficult.** With memory (see Fig. 6), agent pairs more easily find an optimal solution to the task. For the condition where both agents have memory and the listener has partial visibility, message penalties do not seem to decrease performance. The mean over replicas for all tested

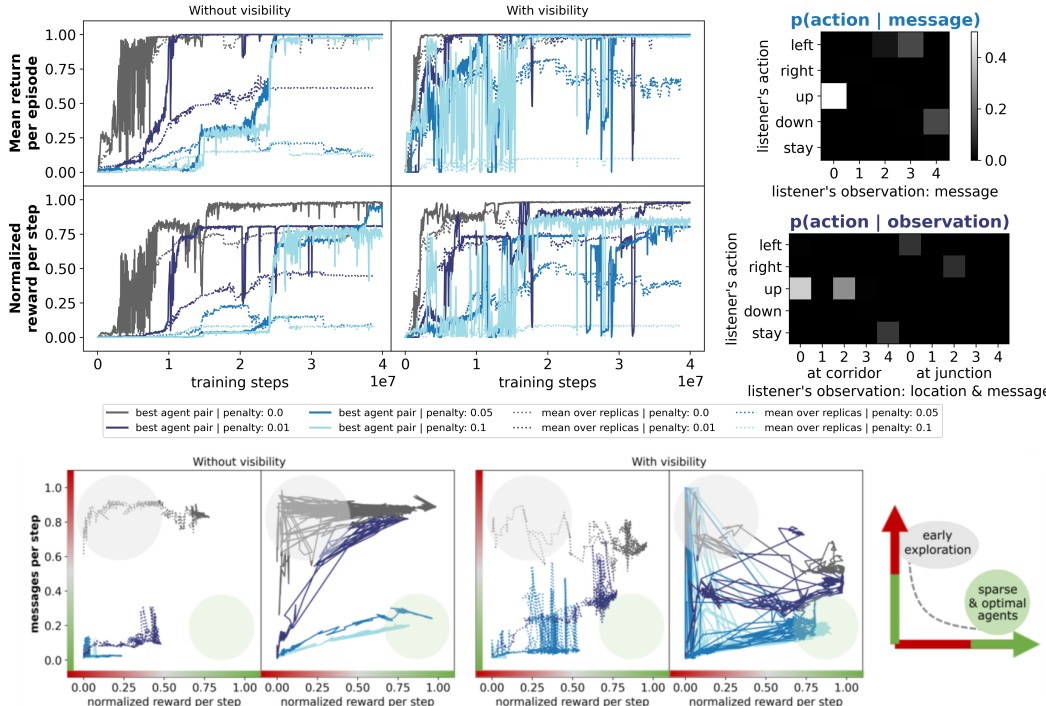

Figure 5: **Agents without memory; speaker experiences a per-message penalty.** When agents converge to a successful communication protocol, the per-message penalty incentivises sparse communication. However, the additional pressure makes learning more difficult—note that overall task performance for blue lines is lower than for the grey line (passive agents with a free communication channel). The immediate cost of non-zero messages stifles early exploration as visible in the plots at the bottom.

penalty values converge to a normalized reward per step similar to the baseline agents with no penalty.

That said, the penalty has little effect on communication sparsity. The communication protocols of two of the best agent pairs are visualized in the heatmaps in Fig. 6. Under the no visibility condition, the visualized speaker uses the null message only about ~10% of the time to communicate a left turn. This is suboptimal—they could be using a free message to communicate 'go straight', which is the most frequent action (~80% of the episode). When the listener has partial visibility, the speaker is more optimal but still sends a non-zero action ~70% of the time.

### A.2.3    SITUATED COMMUNICATION

**The pressure of time in a multi-step interaction can incentivise sparse communication.** In the last set of experiments, we evaluate the impact of *situated* communication on language emergence and find that *situated* communication incentivises conciseness without stifling early exploration. If theoretically optimal, under the no visibility condition an active listener without memory requires 9 messages per episode (1 message per step) and, under partial visibility, they need 2 messages per episode (1 at each junction). The active listener without memory can learn to near optimally solicit information, asking ~9.76 and ~2.06 times per episode under the two visibility conditions, respectively.

The heatmaps in Fig. 7 illustrate the communication protocols of two of the best agent pairs with visibility. Under the partial visibility condition, information solicitation takes place mostly at the junctions, where the acting agent has a choice between two viable environmental actions. Under the no visibility condition (not visualized), the listener queries the speaker for 'move up', 'turn left', and 'turn right' actions, proportionally to their frequency in the task solution. Active listeners without

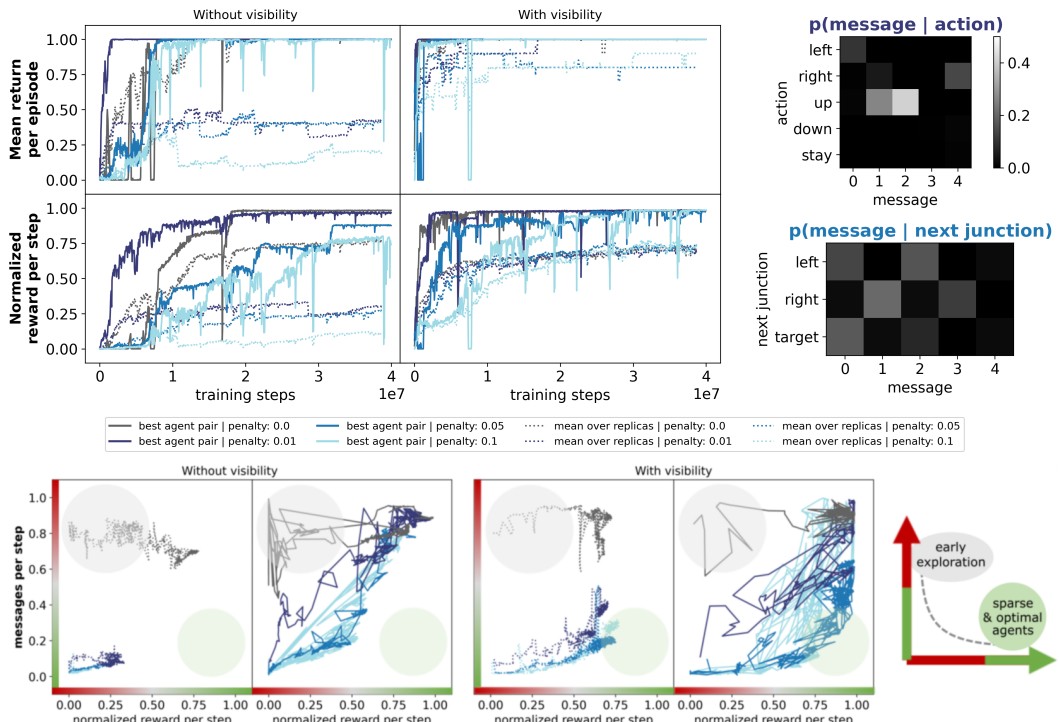

Figure 6: **Agents with memory; speaker experiences a per-message penalty.** With memory, overall performance improves—more agent pairs converge to an optimal solution. However, relative to baseline, agents with a message penalty have more difficulty learning to jointly solve the task. In the heatmaps on the right, we illustrate the communication protocol of the best agent pairs. Note that the penalty is largely ineffective—the agents send many non-zero messages per episode.

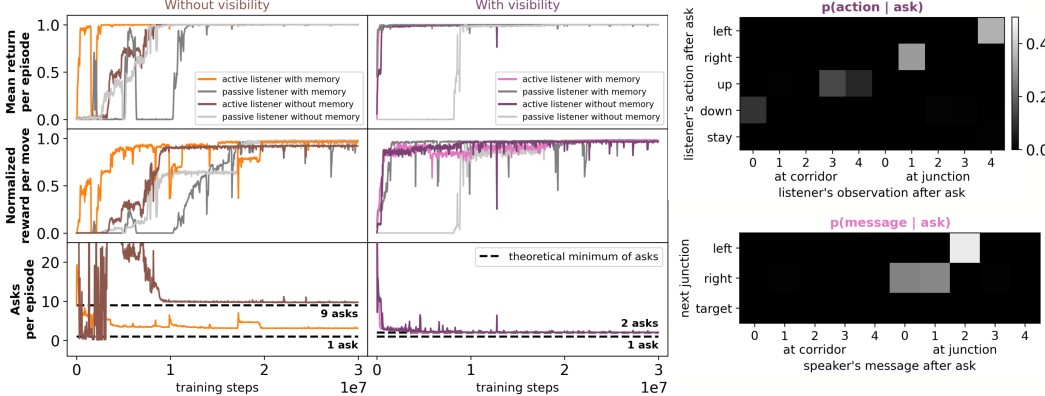

Figure 7: **Best performing pairs of agents with an active listener.** Under all conditions (with/without memory and with/without visibility) agents learn to solve the task via the shortest path. Listeners without memory learn to query the speakers in the optimal number of asks (once per step when the listener has no visibility and once per junction when the listener sees environmental context). Listeners with memory persist to ask for information when it is immediately actionable (instead of once at the beginning of an episode).

memory ask for information when it is immediately actionable as they have no way of reasoning about states in prior timesteps.

**With memory—which gives agents flexibility over message timing—the active listener exhibits a preference for just-in-time communication.** Interestingly, when we test situated communication between agents with memory, agents continue to ask for information at the junctions. This is non-obvious—given memory, the active listener could ask for information at any point in the maze.

In fact, if the agent were to be optimally sparse, they could (1) ask for information only once at the beginning of an episode, (2) receive a message encoding the address of the target, and (3) follow the relevant policy from memory. Instead, the active listener with memory learns sparse communication relative to a passive listener but they do not achieve the theoretically maximum sparsity. An active listener with memory persists to ask for information at the junctions when it is immediately actionable. This result suggests that it may be easier for agents to succeed at the task when they can control the timing of communication.

