# OpenReview forum: "Situated Communication: A Solution to Over-communication between Artificial Agents"
_ICLR.cc/2022/Workshop/EmeCom — EmeCom Workshop at ICLR 2022_

### Official Review · Reviewer_dM1i · 2022-03-22

**Rating:** Accept
**Confidence:** 4

**Review:**

### Summary:
This paper studies emergent communication in dynamic, temporally-extended environments and presents situated communication as a mechanism for incentivizing efficient and concise emergent communication protocols. Situated communication introduces an opportunity cost to communication -- by choosing a communication action (i.e. soliciting information from the sender), the receiver foregoes a movement action in the environment for that time-step -- which translates into an external time pressure on the agents' communication protocol. This method differs from the standard way of assigning a cost to communication actions (per-message penalties), which must be tuned appropriately and may limit the ability of agents to converge on a useful protocol. Empirically, communication is studied in the context of a T-junction navigation task and the performance of situated communication is compared to unsituated communication and unsituated communication with a fixed per-message penalty across multiple environmental/agent conditions (visibility vs no visibility, memory vs. no memory). Results show that situated communication leads to more efficient communication than the baselines, while maintaining similarly high success rates.

### Strengths:
- Situated communication and temporally-extended communication are important topics in the emergent communication literature. This paper is highly relevant to the workshop.
- The discussion of internal vs. external pressures on communication is an interesting one that fits in nicely with the ongoing discourse over aspects of multi-agent learning that might influence communication (e.g. environmental pressures, social pressures, etc). I particularly liked the framing of communication as an opportunity cost.
- The paper is well-written, with lots of detail for both the experimental setup and how results differ across each learning condition (unsituated, unsituated + penalty, situated).

### Questions/Suggestions:
- Emergent communication vs. language development are used loosely / interchangeably throughout.
    - Section 1, P2: For example, the line "However, prior work shows that emergent communication protocols do not share properties of natural language" suggests a difference between emergent protocols and language, but then the learned strategies are referred to as "language emergence" in later sections.
- Figure 2: Which fixed penalty is used here? It seems like if the penalty is high enough, the unsituated + penalty strategy should be the most sparse (or even exchange zero messages) but least effective.
    - It might be worth including multiple curves from Figure 5 so that the results span from no penalty to something more constrained than the situated strategy (or a table if the curves overlap too much).
- Figure 2: It is fascinating that situated communication is more sparse than one message per time-step, even without visibility.
    - The agent has learned that it never needs to ask for information twice in a row, because a junction is always followed by a subsequent step in the same direction. This is more clear with visibility, but seems like it would be harder to discover without visibility.
    - Memory is key here though -- sparsity drops to one message per time-step with no memory and without visibility.
- Is there a sense for how situated communication changes as a function of the number of time-steps to complete the task?
    - If there are very few time-steps, do the agents respond to time pressure by converging more quickly to an effective communication protocol, or does the problem become too difficult to obtain a consistent learning signal?
    - If there are very many time-steps (e.g. no time pressure), do the agents learn to communicate at all, or can the listener agent just learn to search the whole space (since there is no penalty to do so)?
    - For the environment in Figure 1, it looks like it would take ~33 time-steps to search the whole environment with no communication actions (below the 100 time-step limit). Is there be some degenerate case in the space of all policies where the agent follows an action sequence that searches the whole grid within the time limit and never asks for help?
- Are there extensions/cases where over-communication can actually hurt performance?

---

### Official Review · Reviewer_kfhZ · 2022-03-23
**Situated communication review**

**Rating:** Accept
**Confidence:** 4

**Review:**

Summary

This paper introduces non-situated and situated ways of encouraging agents to learn to exchange information concisely. Experiments show that the agents under situated communication can achieve the highest sparsity without negatively affecting communication emergence.

Main review

- This work is novel in that it considers different ways of defining temporal cost. The related elements such as visibility and memory are also taken into account when designing the environment. The corresponding metrics are proposed for evaluating optimality and message sparsity.

- Regarding efficient communication, I would consider two ways, 1) fewer non-zero messages per episode, which is measured in this work 2) more information encoded in each message. These two ways do not necessarily entail each other as the authors mentioned in the appendix. Therefore, measuring how much information each message encodes may be an additional metric for efficient encoding.
The experiments are thorough and designed in accord with the memory and visibility. However, how these two factors contribute to concise communication is not demonstrated in detail. For example, when the receivers can see more pixels in the front, will the message reduce more information of moving forward and keep more information about left, right turn? Additional analysis about what kind of information is compressed and how memory and visibility alone affect the compression will be interesting.

---

### Comment · Program_Chairs · 2022-04-04
**Slight Suggestion for Naming**

The phrase "situated" already has a relatively established meaning in emergent communication literature that is slightly different from what the authors use. From Wagner et al (2003) "Situated simulations place agents in an environment or “artificial world” to which the agents have some causal connection. Just being in an artificial world in which objects can be perceived is not enough for an agent to be classified as situated in this review. To be situated, an agent must also interact in noncommunicative ways with various entities such as food, predators, and other agents and must have outputs that can affect the environment and/or modify its own internal state. On the other hand, in nonsituated simulations an agent’s actions consist solely of sending and receiving signals. Such nonembodied agents do not have noncommunicative interactions with objects or each other beyond being able perhaps to perceive objects or events."

The paper is great and no changes are necessary for the workshop submission, but the authors may consider other possible names for a future conference submission e.g. action-exclusive communication

---

### Comment · Program_Chairs · 2022-04-04
**Best Paper Award**

After going through all accepted papers, the program chairs have decided to award this work the best paper award. We would like to congratulate the authors!

While the topic of sparse / human-like communication has been investigated before, reviewers found this work to be insightful in its approach , thorough in its experiments, and well written. The authors compared against strong baselines and did well to prove the efficacy of their simple but elegant approach. Furthermore, sparse communication and sparse interaction are becoming increasingly interesting for the emergent communication community as efforts to apply emergent communication to real world settings will definitely benefit.

Congratulations! You will be contacted shortly with logistics for the artwork prize :)

---

### Decision · Program_Chairs · 2022-03-25

**Decision:**

Accept

**Comment:**

Reviewers found this work to be novel, having interesting experiments, and well written! We look forward to seeing it at the workshop and discussing different environmental pressures on communication.